# Omental Macrophagic "Crown-like Structures" Are Associated with Poor Prognosis in Advanced-Stage Serous Ovarian Cancer

**Yu-Ling Liang** [1,†], **Chang-Ni Lin** [1,†], **Hsing-Fen Tsai** [1], **Pei-Ying Wu** [1], **Sheng-Hsiang Lin** [2], **Tse-Ming Hong** [2,*] and **Keng-Fu Hsu** [1,2,*]

1   Department of Obstetrics and Gynecology, National Cheng Kung University Hospital, College of Medicine, National Cheng Kung University, Tainan 704, Taiwan; n625526@mail.hosp.ncku.edu.tw (Y.-L.L.); cnlin031@gmail.com (C.-N.L.); tsaihf@mail.ncku.edu.tw (H.-F.T.); anna1002ster@gmail.com (P.-Y.W.)
2   Graduate Institute of Clinical Medicine, National Cheng Kung University Hospital, College of Medicine, National Cheng Kung University, Tainan 704, Taiwan; shlin922@mail.ncku.edu.tw
*   Correspondence: tmhong@mail.ncku.edu.tw (T.-M.H.); d5580@mail.ncku.edu.tw (K.-F.H.); Tel.: +886-6-2353535 (ext. 4259) (T.-M.H.); +886-6-2353535 (ext. 5263) (K.-F.H.); Fax: +886-6-2359885 (T.-M.H.); +886-6-2766185 (K.-F.H.)
†   These authors contributed equally to this work.

**Abstract:** The tumor microenvironment is a well-recognized framework in which immune cells present in the tumor microenvironment promote or inhibit cancer formation and development. A crown-like structure (CLS) has been reported as a dying or dead adipocyte surrounded by a 'crown' of macrophages within adipose tissue, which is a histologic hallmark of the inflammatory process in this tissue. CLSs have also been found to be related to formation, progression and prognosis of some types of cancer. However, the presence of CLSs in the omentum of advanced-stage high-grade serous ovarian carcinoma (HGSOC) has not been thoroughly investigated. By using CD68, a pan-macrophage marker, and CD163, an M2-like polarization macrophage marker, immunohistochemistry (IHC) was performed to identify tumor-associated macrophages (TAMs) and CLSs. This retrospective study analyzed 116 patients with advanced-stage HGSOC who received complete treatment and had available clinical data from July 2008 through December 2016 at National Cheng Kung University Hospital (NCKUH) (Tainan, Taiwan). Based on multivariate Cox regression analysis, patients with omental CD68[+] CLSs had poor OS (median survival: 24 vs. 38 months, $p = 0.001$, hazard ratio (HR): 2.26, 95% confidence interval (CI): 1.41–3.61); patients with omental CD163[+] CLSs also had poor OS (median survival: 22 vs. 36 months, HR: 2.14, 95% CI: 1.33–3.44, $p = 0.002$). Additionally, patients with omental CD68[+] or CD163[+] CLSs showed poor PFS (median survival: 11 vs. 15 months, HR: 2.28, 95% CI: 1.43–3.64, $p = 0.001$; median survival: 11 vs. 15 months, HR: 2.17, 95% CI: 1.35–3.47, respectively, $p = 0.001$). Conversely, the density of CD68[+] or CD163[+] TAMs in ovarian tumors was not associated with patient prognosis in advanced-stage HGSOC in our cohort. In conclusion, we, for the first time, demonstrate that the presence of omental CLSs is associated with poor prognosis in advanced-stage HGSOC.

**Keywords:** crown-like structure (CLS); omentum; tumor microenvironment; advanced-stage high-grade serous ovarian carcinoma (HGSOC)

## 1. Introduction

The tumor microenvironment is a well-recognized framework in which immune cells present promote or inhibit cancer formation and development. A high density of tumor-associated macrophages (TAMs) is reported to be associated with poor prognosis in many human cancer types, including breast, bladder, prostate, head and neck, and endometrial cancers, as well as glioma and melanoma [1–5]. Furthermore, high infiltration of TAMs correlates with better prognosis in some types of cancer, i.e., colorectal cancer [6,7]. However, controversy regarding the impact of TAMs on patient prognosis and clinicopathologic

characteristics remains in gastric cancer [8,9]. In epithelial ovarian cancer (EOC), some studies have shown that a high density of TAMs is significantly associated with worse survival [3,10,11], though others have revealed no difference in survival according to the presence of TAMs [12,13]. Thus, the role of TAMs in ovarian cancer patient survival has yet to be resolved.

EOC is the gynecologic malignancy with the highest mortality due to almost 80% of cases being diagnosed at stage III/IV or advanced-stage disease [14]. In particular, high-grade serous ovarian carcinoma (HGSOC), the most frequent and aggressive form of ovarian cancer, is characterized by the formation of malignant ascites and omental and peritoneal metastases, which results in a disastrous prognosis [15]. The omentum, a visceral adipose tissue in the abdomen formed from a fold of the peritoneal mesothelium, is the most frequent site for metastasis in HGSOC [16]. The omentum contains a high density of lymphoid aggregates or fat-associated lymphoid clusters, which are thought to contribute to peritoneal immunity [17]. The macrophage density in the omentum of ovarian cancer patients was recently shown to increase proportionally with the disease severity score [18], though the specific role of omental macrophages in colonization and disease progression remains to be explored.

Adipocytes are a potential source of energy from stored lipids for metastatic ovarian tumor cells [19]. Metastasizing tumor cells may accumulate in adipose tissue, i.e., the omentum, supporting tumor growth via immunological and metabolic mechanisms [20]. In advanced HGSOC, initial colonization of tumor cells triggers an increase in lymphoid aggregate size and number, mainly through macrophage recruitment from the peritoneal cavity [17]. However, adipocytes are directly involved in tumor initiation, progression, invasion, and metastasis in breast cancer [21]. In general, the interaction between adipocytes and macrophages appears to play some role in cancer progression [22].

Crown-like structures (CLSs), reported as dying or dead adipocytes surrounded by a 'crown' of macrophages within adipose tissue, are a histologic hallmark of the inflammatory process in adipose tissue [23–25]. As inflammation is a critical component of tumor progression [26], CLSs have also been found to be related to the formation, progression and prognosis of some types of cancer. In breast cancer, a high density of CLS in patients with benign disease was correlated with significantly increased odds ratios for breast cancer development, implying a link between CLS and breast carcinogenesis [27–30]. In early-stage squamous cell carcinoma of the tongue, Iyengar et al. reported that CLS presence is associated with worse disease-specific survival and overall survival [31], and according to Gucalp et al., CLSs in periprostatic fat tissue are associated with high-grade prostate cancer [32]. In a nonalcoholic steatohepatitis mouse model, hepatic CLSs, in which $CD11c^+$ macrophages surround dead/dying hepatocytes with large lipid droplets, drives hepatocyte death-triggered liver fibrosis; this would predispose patients to the development of hepatocellular carcinoma [33]. Although the omentum is the most frequent metastatic site in advanced-stage HGSOC, the presence of CLSs in the omentum has not been thoroughly investigated.

In this study, we aimed to evaluate and analyze the macrophage CLS status in omental adipose tissue, as well as the association of survival in patients with advanced-stage HGSOC. As few studies have described the role of omental TAMs in advanced HGSOC, we also examined correlations between omental TAMs and prognosis in these patients.

## 2. Materials and Methods

### 2.1. Patient Enrollment

From July 2008 to December 2016, patients who underwent surgery for ovarian cancer at National Cheng Kung University Hospital (NCKUH), with a final pathologic report that, according to the 2014 International Federation of Gynecology and Obstetrics (FIGO), defined the stage III/IV disease and high-grade serous ovarian cancer, were consecutively enrolled. The protocol was approved by the Institutional Review Board of NCKUH (No: B-ER-108-434), and informed consent was obtained from all subjects.

Patient clinical data were collected, including FIGO stage (2014), body mass index (BMI), clinicopathologic characteristics, treatment modalities, recurrence status and survival status. Optimal cytoreduction was considered when the maximum diameter of residual disease was less than 1 centimeter. Survival time was calculated from the date of surgery. Overall survival (OS) was determined based on the date of death or the date of last contact for living patients. Progression-free survival (PFS) was determined based on the date of first progression or death, whichever occurred first, or the date of last contact for living patients with or without recurrent disease. Disease progression was based on Response Evaluation Criteria in Solid Tumors (RECIST) or serially increasing CA125 levels or any clinical or radiographic evidence of new lesions as either local/regional relapse or distant metastasis [34,35]. All patients received adjuvant platinum-based chemotherapy, except those whose performance status was too poor to receive it. Cases with disease progression or disease recurrence <6 months after discontinuing chemotherapy were defined as chemoresistant; cases without recurrence or with recurrence ≥6 months after discontinuing chemotherapy were defined as chemosensitive [36]. All procedures were carried out in accordance with approved NCKUH guidelines. The exclusion criteria were as follows: no histologic confirmation of the diagnosis and inadequate data in the medical record.

### 2.2. Immunohistochemistry and TAM, CLS Assessment

CD68, a pan-macrophage marker, and CD163, an M2-like polarization macrophage marker, were used for TAM and CLSs identification [1,11,37,38]. Immunohistochemistry (IHC) was performed using a conventional method described previously [39]. Formalin-fixed paraffin-embedded tissue samples were obtained and stained with a GST control antibody or anti-CD68 (M0814; Dako, Carpinteria, CA, USA) or anti-CD163 (TA506388; Origene, Rockville, MD, USA) antibodies. H&E staining was performed to verify the tumor cell composition of the paraffin sections, and serial sections of tissues were used for IHC to determine CD68 and CD163 expression. Briefly, the sections were serially dewaxed, rehydrated, and processed for antigen retrieval by heating with 10 mM sodium citrate (pH 6.0) for 20 min. After blocking endogenous peroxidases with 3% hydrogen peroxide, the tumor sections were incubated with the primary antibody (1:300 for CD68, 1:500 for CD163) overnight at 4 °C. Bound primary antibodies were detected using the LSAB kit (Dako, Carpinteria, CA, USA), and the slides were counterstained with hematoxylin.

CLS was defined as a structure in the omentum composed of adipocytes completely surrounded by CD68$^+$ or CD163$^+$ macrophages, as shown in Figure 1B or Figure S4B. To determine the number of CLSs, five fields that contained the most CD68$^+$ or CD163$^+$ CLSs were selected under 200× magnification, and the number of CLSs was calculated with the color deconvolution plug-in ImageJ software, as illustrated in Figure 1C or Figure S4C. The TAM density in the omentum (om-TAM) and in the primary ovarian tumor (ov-TAM) were determined based on the area with the greatest concentration of CD68- or CD163-stained macrophages. Five fields (four quadrants and a central area, each field contained 0.04 mm$^2$) were selected, and CD68$^+$ or CD163$^+$ macrophages were counted under 200× magnification (Figures 1G and S4G). The average number of CD68$^+$ or CD163$^+$ macrophages per field was recorded as the TAM density (cells/0.04 mm$^2$). The plug-in ImageJ software was used to assist in calculating the number of om-TAMs (Figures 1F and S4F) and ov-TAMs (Figures 1I and S4I). Moreover, the omentum specimens containing different CLS amounts were shown in Figure S5.

The cutoff values for high/low density om-TAMs, ov-TAMs, and number of CLSs were determined by the median value. Because the median values of CD68$^+$ CLSs and CD163$^+$ CLSs were both zero, patients with omental CLSs were separated into two groups: CLS-present ($n \geq 1$) and CLS-absent ($n = 0$). The median values of CD68$^+$ om-TAMs, CD163$^+$ om-TAMs, CD68$^+$ ov-TAMs, and CD163$^+$ ov-TAMs were 29, 26, 38, 26 cells/0.04 mm$^2$, respectively. The median values of the ratio of CD68$^+$/CD163$^+$ om-TAMs and the ratio of CD68$^+$/CD163$^+$ ov-TAMs were 0.9 and 1.77, respectively.

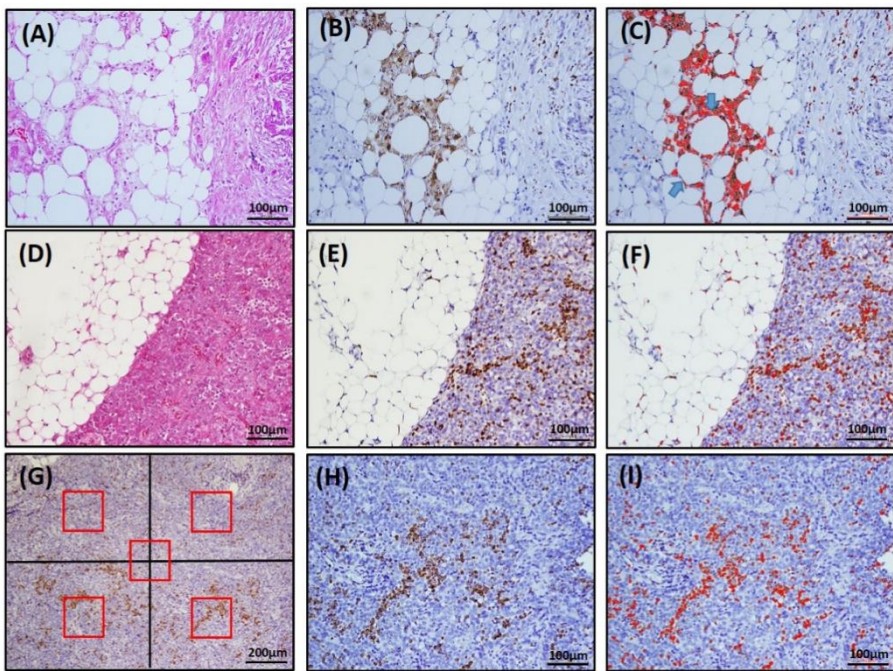

**Figure 1.** Representative image of CLS and TAMs by immunohistochemical staining for CD68 and ImageJ software-assisted images. (**A**,**D**,**G**), H&E staining; (**B**,**E**,**H**), IHC CD68 staining; (**C**,**F**,**I**), ImageJ images. In (**C**), note that there are two adipocytes completely surrounded by CD68-positive macrophages (arrows), counted as two CD68⁺ CLSs. (**G**) For TAM density, the area with the greatest concentration of CD68-stained TAMs was identified. Then, five fields (red box, four quadrants and a central area, each field contained 0.04 mm$^2$) under 100× magnification were selected, and the number of CD68⁺ macrophages was counted under 200× magnification with the assistance of ImageJ software (**I**).

## 3. Statistical Analyses

Statistical analysis of OS and PFS was performed using SPSS software (Version 26.0; IBM Corp, Armonk, NY, USA) and GraphPad 5 (GraphPad Software, Inc., San Diego, CA, USA). Multicollinearity was assessed by using the variance inflation factor (VIF) [40] within SAS statistical software (version 9.4, SAS Institute Inc., Cary, NC, USA); variables with a VIF value over 10 were considered to have a high degree of multicollinearity in the model. Survival curves were generated using the Kaplan–Meier method, and differences between survivors were assessed by the log-rank test. Univariate and multivariate survival analyses were performed using Cox proportional hazards models to identify prognostic factors, and factors that were prognostically relevant in univariate analysis were included in multivariate Cox analysis. Effects are expressed as hazard ratios (HRs) with 95% confidence intervals (CIs). Linear regression was employed to model the relationship between various TAM densities and CLSs. A $p$ value < 0.05 was considered statistically significant.

## 4. Results

### 4.1. Patient Characteristics

From July 2008 to December 2016, 293 EOC patients were treated at NCKUH. Among them, 118 had stage III disease, and 32 had stage IV disease. After excluding ineligible cases, 116 patients with FIGO stage III/IV HGSOC were enrolled in this study. Table 1 shows the association between clinical parameters and omental TAM and CLS in 116 advanced-stage HGSOC patients. The mean age at diagnosis was 57.1 years (range 23–88). Among them, 5 patients received neoadjuvant chemotherapy; 103 (88.75%) had stage III disease, and 13 had stage IV disease (11.3%). There were 71 patients (61%) with optimal debulking (tumor < 1 cm) surgery. Most patients (93/116, 88%) had massive ascites (mean: 2235 cc, range: 0–8200 cc). A total of 112 patients (96.5%) received platinum-based adjuvant

chemotherapy as the first-line chemotherapy, 60% of whom were sensitive. At the time of the last follow-up, 19 patients (16.3%) were alive without evidence of disease, 10 (8.6%) were alive with disease, 83 (71.6%) had died from the disease, and 4 (3.4%) had died from other causes. The mean follow-up time was 44 months (range: 0 to 140 months). The median OS and PFS were 35 months and 13 months, respectively.

In our cohort, we observed that patients with a high density of omental CD68$^+$ om-TAMs were younger than those with a low omental density of CD68$^+$ om-TAMs (54.2 vs. 60.1, *p* = 0.01, Table 1). Patients with a low density of omental CD68$^+$ and CD163$^+$ om-TAMs had a higher BMI (24 vs. 21, p=0.01; 24 vs. 21, *p* = 0.01, Table 1), and omental CD68$^+$ or CD163$^+$ CLSs were associated with higher BMI (27 vs. 21, *p* = 0.01; 27 vs. 21, *p* = 0.01, Table 1). Patients with a high density of omental CD68$^+$, CD163$^+$ om-TAMs were more sensitive to chemotherapy than those with a low density of CD68$^+$, CD163$^+$ om-TAMs (*p* = 0.002, *p* = 0.006, respectively, Table 1). In addition, omental CD68$^+$ or CD163$^+$ CLSs were associated with resistance to chemotherapy (*p* = 0.006, *p* = 0.006, respectively, Table 1). The association between clinical parameters and primary ovarian tumor TAMs in 116 advanced-stage HGSOC ECO patients is shown in Supplementary Table S1. Patients with a high density of ovarian tumor CD163$^+$ ov-TAMs had a higher BMI (24 vs. 22, *p* = 0.03, Table S1). In contrast, neither CD68$^+$ ov-TAMs, CD163$^+$ ov-TAMs nor the ratio of CD68$^+$/CD163$^+$ ov-TAMs was associated with age, FIGO stage, residual tumor or chemosensitivity (Table S1).

*4.2. The Number of Omental CD68$^+$ Clss Had a Strongly Positive Correlation with Omental CD163$^+$ CLS; Density of CD68$^+$ om-TAMs Had a Strong Positive Correlation with the Density of CD163$^+$ om-TAMs*

We performed regression analysis to understand the relationship between omental CLSs and TAMs in advanced-stage HGSOC patients. As depicted in Figure 2, there were strongly positive correlations between the number of omental CD68$^+$ CLSs and CD163$^+$ CLSs (r = 0.77, *p* < 0.0001, Figure S1A) and the density of CD68$^+$ om-TAMs and CD163$^+$ om-TAMs (r = 0.78, *p* < 0.0001, Figure S1B). However, little correlation between the density of CD68$^+$ om-TAMs and the number of CD68$^+$ CLSs (r = −0.23, *p* = 0.02, Figure S1C), the density of CD 163$^+$ om-TAMs and the number of CD163$^+$ CLSs was noted (r =−0.271, *p* = 0.004, Figure S1D), though both *p* values were less than 0.05.

*4.3. Advanced-Stage HGSOC with Omental CD68$^+$ or CD163$^+$ CLSs Is Associated with Poor Prognosis*

To further understand the possible prognostic role of ov-TAM, om-TAM and CLSs, as well as other clinical parameters, i.e., age, FIGO stage, and residual tumor status, a Cox regression analysis for OS and PFS was performed, as shown in Tables 2 and 3. Based on the median as the cutoff value, we grouped all 116 patients into two groups with the presence or absence of omental CD68$^+$ or CD163$^+$ CLSs. Patients with omental CD68$^+$ CLSs had poor OS (median survival: 24 vs. 38 months, *p* = 0.001, HR: 2.26, 95% CI: 1.41–3.61, Figure 2A, Table 2). Additionally, patients with omental CD163$^+$ CLSs had poor OS (median survival: 22 vs. 36 months, HR: 2.14, CI: 1.33–3.44, *p* = 0.002, Figure 2B, Table 2). Omental CD68$^+$ or CD163$^+$ CLSs were also related to poor PFS (median survival: 11 vs. 15 months, HR: 2.28, CI: 1.43–3.64, *p* = 0.001; median survival: 11 vs. 15 months, HR: 2.17, CI: 1.35–3.47, respectively, *p* = 0.001, Figure 2C,D, Table 3).

**Table 1.** Association between clinical parameters and omental TAM and CLS in 116 advanced stage serous ovarian cancer patients.

| Variable | | CD68$^+$ om-TAM | | | CD163$^+$ om-TAM | | | Ratio of CD68$^+$/CD163$^+$ om-TAM | | | CD68$^+$ CLS | | | CD163$^+$ CLS | | |
|---|---|---|---|---|---|---|---|---|---|---|---|---|---|---|---|---|
| | Total | High | Low | *p* Value | High | Low | *p* Value | High | Low | *p* Value | Present | Absent | *p* value | Present | Absent | *p* Value |
| | *n*= 116 | (*n* = 59) | (*n* = 57) | | (*n* = 63) | (*n* = 53) | | (*n* = 59) | (*n* = 57) | | (*n* = 31) | (*n* = 85) | | (*n* = 30) | (*n* = 86) | |
| Age (y/o) Mean (range) | 57.1 (23~88) | 54.2 (23~88) | 60.1 (36~82) | **0.01** | 55.2 (23~88) | 59.2 (36~82) | 0.09 | 56.6 (30~88) | 57.5 (23~82) | 0.72 | 57.8 (30~82) | 57 (23~88) | 0.71 | 57.8 (30~82) | 56.7 (23~88) | 0.7 |
| BMI Mean (range) | 23 (15~34) | 21 (15~26) | 24 (18~34) | **0.01** | 21 (15~34) | 24 (18~34) | **0.01** | 23 (15~34) | 23 (15~31) | 0.63 | 27 (25~34) | 21 (15~25) | **0.01** | 27 (25~34) | 21 (15~27) | **0.01** |
| FIGO Stage (%) | | | | 1 | | | 0.77 | | | 0.39 | | | 0.18 | | | 0.18 |
| IIIA | 4 (3) | 0 (0) | 4 (7) | | 0 (0) | 4 (8) | | 0 (9) | 4 (7) | | 0 | 4 (5) | | 0 (0) | 4 (5) | |
| IIIB | 17 (15) | 11 (19) | 6 (10) | | 10 (16) | 7 (12) | | 9 (15) | 8 (14) | | 4 (13) | 13 (15) | | 4 (13) | 13 (15) | |
| IIIC | 82 (71) | 41 (70) | 41 (72) | | 45 (71) | 37 (70) | | 45 (77) | 37 (65) | | 26 (84) | 56 (66) | | 25 (84) | 57 (66) | |
| IVA | 9 (8) | 5 (8) | 4 (7) | | 5 (8) | 4 (8) | | 3 (5) | 6 (11) | | 1 (3) | 8 (9) | | 1 (3) | 8 (9) | |
| IVB | 4 (3) | 2 (3) | 2 (4) | | 3 (5) | 1 (2) | | 2 (3) | 2 (3) | | 0 (0) | 4 (5) | | 0 (0) | 4 (5) | |
| Residual Disease (%) | | | | 0.34 | | | 0.7 | | | 0.71 | | | **0.05** | | | 0.08 |
| Optimal (≤1 cm) | 71 (61) | 39 (66) | 32 (56) | | 40 (64) | 31 (58) | | 35 (59) | 36 (63) | | 14 (45) | 57 (67) | | 14 (47) | 57 (66) | |
| not-optimal (>1 cm) | 45 (39) | 20 (34) | 25 (44) | | 23 (36) | 22 (42) | | 24 (41) | 21 (37) | | 17 (55) | 28 (33) | | 16 (53) | 29 (34) | |
| Chemotherapy (%) | | | | **0.002** | | | **0.006** | | | 0.44 | | | **0.006** | | | **0.006** |
| sensitive | 70 (60) | 45 (76) | 25 (44) | | 46 (73) | 24 (45) | | 38 (65) | 32 (56) | | 11 (35) | 59 (70) | | 11 (37) | 59 (69) | |
| resistant | 42 (36) | 14 (24) | 28 (49) | | 16 (25) | 26 (49) | | 19 (32) | 23 (40) | | 17 (55) | 25 (29) | | 17 (57) | 25 (29) | |
| No chemotherapy | 4 (4) | 0 (0) | 4 (7) | | 1 (2) | 3 (6) | | 2 (3) | 2 (4) | | 3 (10) | 1 (1) | | 2 (6) | 2 (2) | |

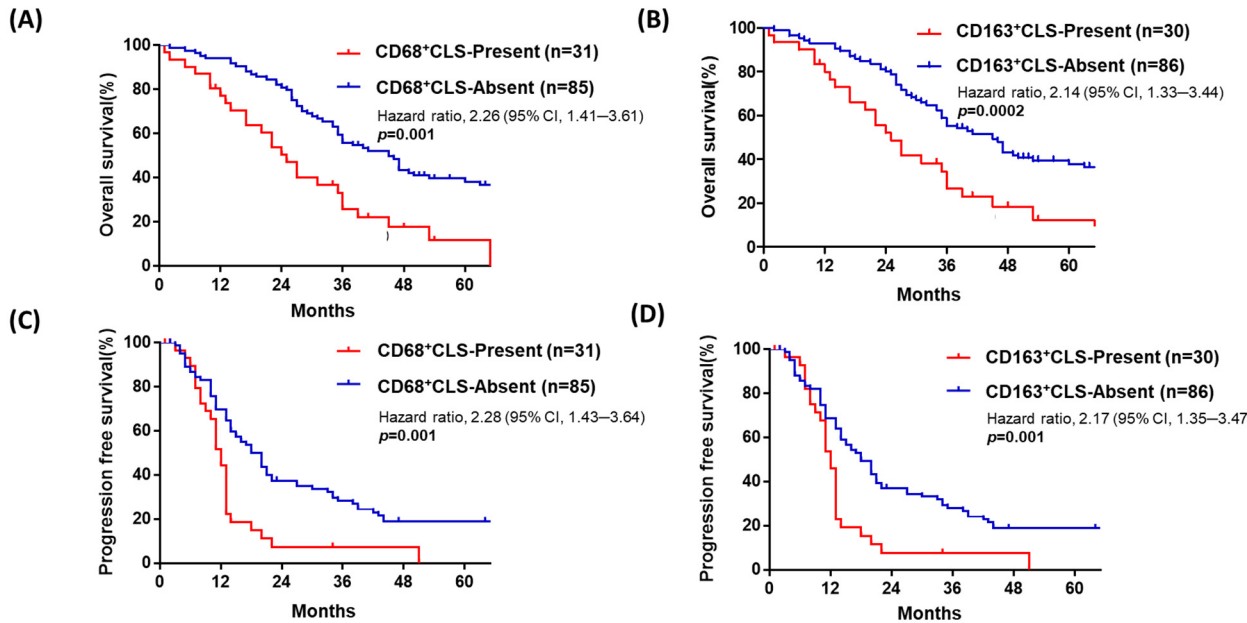

**Figure 2.** Patients with advanced-stage HGSOC with omental CD68[+] or CD163[+] CLSs have poorer OS and PFS than those without omental CLSs. In advanced-stage HGSOC ($n$ = 116), patients with omental CD68[+] CLSs (no. $\geq$ 1) (**A**) or omental CD163[+] CLSs (no. $\geq$ 1) (**B**) had worse OS (median survival: 24 vs. 38 months; 22 vs. 36 months, respectively, $p$ = 0.001; $p$ = 0.002). Patients with omental CD68[+] (**C**) or CD163[+] (**D**) CLSs had poor PFS (median survival: 11 vs. 15 months; 11 vs. 15 months, respectively $p$ = 0.001, 0.001).

**Table 2.** Univariate and multivariate Cox proportional hazards regression model for overall survival ($n$= 116).

| Variable | Univariate Analysis | | Multivariate Analysis | | | |
| | | | Model 1 (CD68) [a] | | Model 2 (CD163) [b] | |
| | HR (95% CI) | $p$ Value | HR (95% CI) | $p$ Value | HR (95% CI) | $p$ Value |
|---|---|---|---|---|---|---|
| Age | 1.03 (1.02–1.05) | **<0.001** | 1.03 (1.01–1.05) | **0.004** | 1.03 (1.01–1.05) | **0.003** |
| FIGO Stage (Stage III vs. Stage IV) | 1.13 (0.58–2.19) | 0.727 | | | | |
| Residual disease (suboptimal debulking vs. optimal debulking) | 1.65 (1.07–2.56) | **0.025** | 1.25 (0.79–1.99) | 0.346 | 1.26 (0.79–2.02) | 0.328 |
| ov-TAM (High-CD68[+] vs. Low-CD68[+]) | 1.02 (0.67–1.55) | 0.933 | | | | |
| ov-TAM (High-CD163[+] vs. Low-CD163[+]) | 1.20 (0.79–1.83) | 0.403 | | | | |
| Ratio of ov-TAM (High-CD68/CD163 vs. Low-CD68/CD163 ) | 1.32 (0.87–2.02) | 0.196 | | | | |
| om-TAM (Low-CD68[+] vs. High-CD68[+]) | 1.63 (1.07–2.49) | **0.023** | 1.07 (0.66–1.73) | 0.791 | | |
| om-TAM (Low-CD163[+] vs. High-CD163[+]) | 1.43 (0.94–2.18) | 0.098 | | | 1.01 (0.62–1.64) | 0.982 |
| Ratio of om-TAM (Low-CD68/CD163 vs. High-CD68/CD163 ) | 1.16 (0.76–1.77) | 0.506 | | | | |
| CD68[+] CLS (present vs. absent) | 2.26 (1.41–3.61) | **0.001** | 2.03 (1.19–3.44) | **0.009** | | |
| CD163[+] CLS (present vs. absent) | 2.14 (1.33–3.44) | **0.002** | | | 1.98 (1.14–3.41) | **0.015** |

CI, confidence interval; HR, hazard ratio; CD68[+], CD68-positive; CD163[+], CD163-positive; CLS, crown-like structure; TAM, tumor-associated macrophage. [a] as the focus on CD68 analysis; [b] as the focus on CD163 analysis.

**Table 3.** Univariate and multivariate Cox proportional hazards regression model for progression-free survival (*n*= 116).

| | Univariate Analysis | | Multivariate Analysis | | | |
| | | | Model 1 (CD68) [a] | | Model 2 (CD163) [b] | |
| Variable | HR (95% CI) | *p* Value | HR (95% CI) | *p* Value | HR (95% CI) | *p* Value |
|---|---|---|---|---|---|---|
| Age | 1.02 (1.01–1.04) | **0.01** | 1.01 (0.99–1.03) | 0.22 | 1.01 (0.99–1.03) | 0.22 |
| FIGO Stage (Stage III vs. Stage IV) | 1.20 (0.64–2.25) | 0.58 | | | | |
| Residual disease (suboptimal debulking vs. optimal debulking) | 2.43 (1.59–3.71) | **<0.001** | 1.97 (1.23–3.16) | **0.005** | 1.95 (1.21–3.15) | **0.007** |
| ov-TAM (High-CD68$^+$ vs. Low-CD68$^+$) | 1.17 (0.78–1.74) | 0.452 | | | | |
| ov-TAM (High-CD163$^+$ vs. Low-CD163$^+$) | 1.23 (0.83–1.84) | 0.305 | | | | |
| Ratio of ov-TAM (High-CD68/CD163 vs. Low-CD68/CD163 ) | 1.19 (0.80–1.78) | 0.392 | | | | |
| om-TAM (High-CD68$^+$ vs. Low-CD68$^+$) | 1.29 (0.86–1.93) | 0.225 | 1.18 (0.72–1.89) | 0.522 | | |
| om-TAM (High-CD163$^+$ vs. Low-CD163$^+$) | 1.13 (0.75–1.70) | 0.547 | | | 1.19 (0.74–1.92) | 0.486 |
| Ratio of om-TAM (High-CD68/CD163 vs. Low-CD68/CD163) | 1.24 (0.83–1.86) | 0.299 | | | | |
| CD68$^+$ CLS (present vs. absent) | 2.28 (1.43–3.64) | **0.001** | 2.11 (1.23–3.64) | **0.007** | | |
| CD163$^+$ CLS (present vs. absent) | 2.17 (1.35–3.47) | **0.001** | | | 2.02 (1.17–3.48) | **0.012** |

CI, confidence interval; HR, hazard ratio; CD68$^+$, CD68-positive; CD163$^+$, CD163-positive; CLS, crown-like structure; TAM, tumor-associated macrophage. [a] as the focus on CD68 analysis; [b] as the focus on CD163 analysis.

For multivariate Cox regression analysis, we first assessed multicollinearity by the variance inflation factor (VIF) and observed a high degree of multicollinearity for CD68$^+$ CLSs and CD163$^+$ CLSs (Figure S2) for OS (Figure S2A) and PFS (Figure S2B). Therefore, we only included the results of CD68-stained CLSs in model 1 and the results of CD163-stained CLSs in model 2 to perform multivariate analysis. The results showed that age (HR: 1.03, CI: 1.01–1.05, *p* = 0.004) and the presence of CD68$^+$ CLSs (HR: 2.03, CI: 1.19–3.44, *p* = 0.009) in model 1 and the presence of CD163$^+$ CLSs (HR: 1.98, CI: 1.14–3.41, *p* = 0.015) in model 2 were significantly associated with worse OS (Table 2). For PFS analysis, residual tumor disease (HR: 1.97, CI: 1.23–3.16, *p* = 0.005) and the presence of CD68$^+$ CLSs (HR: 2.11, CI: 1.23–3.64, *p* = 0.007) in model 1 and the presence of CD163$^+$ CLSs (HR: 2.02, CI: 1.17–3.48, *p* = 0.012) in model 2 were significantly associated with worse PFS. However, the density of omental om-TAMs was not associated with OS (Table 2) or PFS (Table 3) in multivariate model 1 or model 2 analysis.

*4.4. Patients with Advanced-Stage HGSOC with a Low Density of Omental CD68$^+$ or CD163$^+$ om-TAMs Have Poor OS, though the Ovarian Tumor Density of CD68$^+$ or CD163$^+$ ov-TAMs Is Not Associated with Patient Prognosis*

Patients with a low density of omental CD68$^+$ or CD163$^+$ om-TAMs had poor OS (median survival: 27 vs. 47 months, HR: 1.63, CI: 1.07–2.49, *p* = 0.023; 26 vs. 46 months, HR: 1.43, CI: 0.94–2.18, *p* = 0.098) (Figure 3A,B), but the density of omental CD68$^+$ or CD163$^+$ om-TAMs was not associated with PFS (median survival: 13 vs. 20 months; 13 vs. 18 months, *p* = 0.225, *p* = 0.547) (Figure 3D,E). Furthermore, the ratio of omental CD68$^+$/ CD163$^+$ om-TAM was not associated with OS or PFS (*p* = 0.50, *p* = 0.299, Figure 3C,F). Overall, we found that neither ovarian tumor densities of ov-CD68$^+$ or CD163$^+$ ov-TAMs nor the ratio of ovarian CD68$^+$/CD163$^+$ ov-TAMs was significantly associated with patient OS or PFS (Tables 2 and 3, Figure S3).

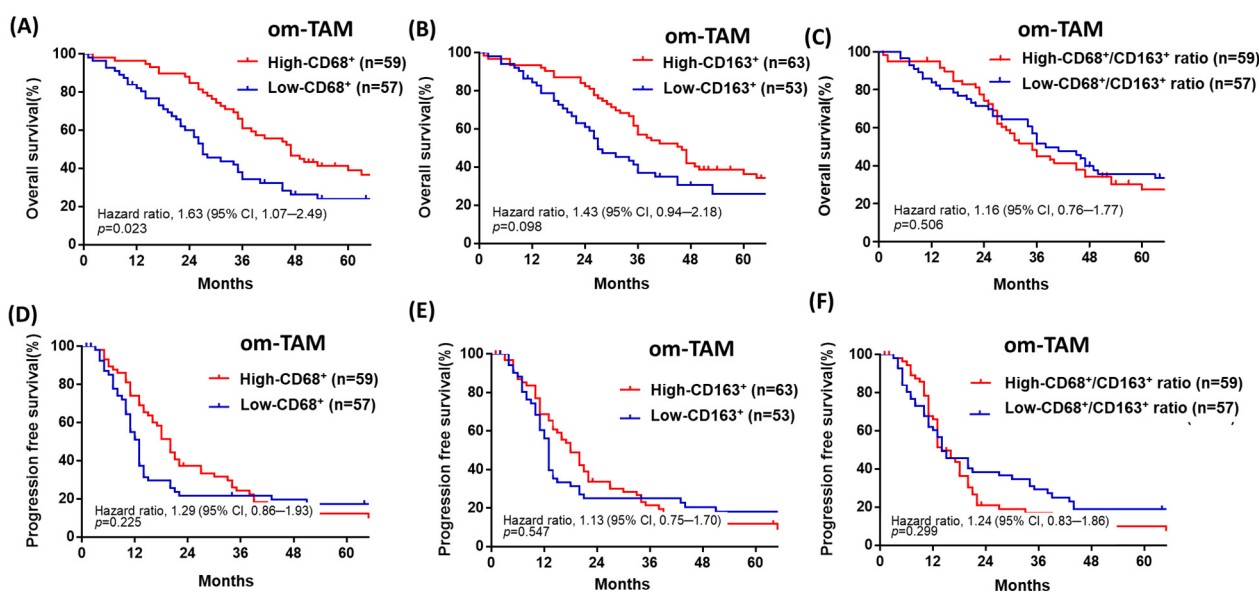

**Figure 3.** Patients with advanced-stage HGSOC with a low density of omental CD68+ or CD163+ om-TAMs had a poor prognosis in terms of OS. (**A,B**) Patients with a low density of omental CD68+ or CD163+ om-TAMs had poor OS (median survival: 27 vs. 47 months; 26 vs. 46 months, *p* = 0.023; *p* = 0.098). (**C,D**) The density of omental CD68+ or CD163+ om-TAMs was not associated with PFS (median survival: 13 vs. 20 months; 13 vs. 18 months, *p* = 0.225, *p* = 0.547). (**E,F**) The ratio of omental CD68+ / CD163+ om-TAM was not associated with OS or PFS (*p* = 0.50; *p* = 0.299).

## 5. Discussion

In this study, we found that the presence of omental CD68+ or CD163+ CLSs is associated with poor prognosis in advanced-stage HGSOC, though the density of ovarian tumor TAMs was not associated with patient survival in our cohort. In the omentum of advanced-stage HGSOC, there was little correlation between CLS count and TAM density, suggesting the independence of these two processes.

CLSs, consisting of dead or dying adipocytes surrounded by macrophages, are a marker of adipose tissue inflammation [23–25], and they have been reported to be associated with poor prognosis in breast cancer [27–30]. However, Maliniak et al. indicated that CLS number was not associated with OS or PFS in stage I–III breast cancer [41], which may be related to differences in methodology in CLS assessment. In our study of advanced-stage HGSOC, we examined CD68 or CD163 for CLS and TAM determination. CLSs were defined as adipocytes completely surrounded by CD68+ or CD163+ macrophages, and the area with the greatest concentration of CD68+ or CD163+ CLSs was selected and counted. In regression analysis, omental CD68+ om-CLS correlated significantly positively with CD163+ om-CLSs (r = 0.77, *p* < 0.0001), as did the density of CD68+ om-TAMs with the omental density of CD163+ om-TAMs (r = 0.78, *p* < 0.0001) (Figure S1). The results suggest high homogeneity of omental CD68- and CD163-stained macrophages in advanced-stage HGSOC. Reinartz et al. reported similar results. Through a comprehensive genome-wide expression profile of TAMs in ascites in patients with HGSOC, they found that TAMs exhibit an M1/M2 hybrid phenotype because they express both M2, such as IL-10 and CD163, and M1, such as CD86 and TNF-α, markers [42].

Adipocytes are the most abundant cellular component of omental and peritoneal adipose tissue [17]. Cancer-associated adipocytes, CAAs, exhibit an inflammatory phenotype, releasing cytokine/chemokine adipokines, i.e., leptin, adiponectin, interleukin-6 (IL-6), and IL-8 that contribute to ovarian cancer cell metastatic colonization [19,20]. In particular, adipocytes promote ovarian cancer metastasis and are associated with chemoresistance [19,43,44]. CAAs release fatty acids through lipolysis, which are then transferred to cancer cells and used for energy production via β-oxidation. The abundant availability

of lipids from adipocytes in the tumor microenvironment supports uncontrolled growth and tumor progression [20].

Adipocytes contribute to adipose tissue macrophage differentiation and functions change through paracrine regulation [45]. Macrophages also play key roles in lipid metabolism, disruption of which can result in pathologies such as atherosclerosis [46]. The tumor microenvironment (TME) has also been characterized as a lipid-rich context, and how lipids in the TME affect macrophage pro- or antitumor function is of interest. In a mouse melanoma model, TAMs take up oxidized LDL through the scavenger receptor CD36, contributing to protumor function [47]. Similarly, TAMs express elevated levels of CD36, accumulate lipids, and utilize fatty acid oxidation instead of glycolysis for energy [48]. In a thyroid carcinoma and neuroblastoma coculture model, tumor cells stimulate lipid biosynthesis in macrophages to induce protumoral factor cytokine production and reactive oxygen species responses [49]. Tumor-derived exosomes have received widespread attention for their role in cancer progression and metastasis [50], and CLS-associated macrophages may secrete exosomes, facilitating direct communication between CLSs and tumors [37]. In our cohort, patients with omental CD68[+] om-CLS or CD163[+] om-CLSs showed chemoresistance, as well as poor prognosis compared with those without om-CLSs. The protumor effect of the presence of om-CLSs may be due to the contribution of CAAs, the high lipid metabolism in omental macrophages, or the communication between CLS-associated macrophages and tumor cells. Further studies will be needed to address this.

Accumulating evidence indicates that immune cells play an important role in cancer progression and act as prognostic markers. Some studies demonstrate high M2-like TAM (CD163[+]) levels in primary ovarian cancer tumors, and a high proportion of the CD163[+]/CD68[+] TAM phenotype, is associated with poor PFS [11,51,52]. Nevertheless, we did not observe that ovarian tumor TAMs are associated with survival in our cohort of advanced HGSOC patients. The factors related to these heterogeneous findings probably involve tissue selection, the diversity of macrophages and the experimental approach used to investigate their correlation with patient prognosis. Additionally, different studies evaluating macrophage density or numbers may have considered only macrophages in some specific regions (i.e., peritumor or perivascular), whereas others may have used activation/polarization markers that capture only a fraction of the whole population. Therefore, based on the reasons mentioned above, it seems logical that the results for TAMs may be well associated with distinct prognostic behaviors [53].

There were four patients with stage IIIa disease in our cohort. Although no pathologic omental tumor metastasis occurred, we still assessed the number of omental om-TAMs and CLSs. As advanced-stage HGSOC is a disease with transcoelomic metastasis [54], the whole peritoneal cavity is similar to the TME, with cytokines, chemokines, and growth factors, which may influence TAM migration to the omentum in primary ovarian cancer [18].

Some studies have shown that obesity measured by body mass index (BMI) has a positive association with CLSs in breast cancer patients [27,29,41]. Nonetheless, CLSs are reportedly present in up to one-third of women with a lean status (BMI < 25 kg/m$^2$) [55], suggesting that factors other than obesity contribute to the formation of CLSs; however, we did find that the presence of CLSs was associated with high BMI in our cohort (Table 1). Moreover, patients with advanced-stage serous ovarian cancer often have massive ascites and edema, as in one of our patients (mean volume ascites, 2235 cc). Therefore, it is difficult to accurately determine BMI in these patients.

In conclusion, we, for the first time, demonstrated that the presence of omental CLSs is associated with poorer prognosis than the absence of omental CLSs in advanced-stage HGSOC. In contrast, due to the complexity of TAM calculations, the presence or absence of omental CLSs is a more facile approach to studying patient prognosis. Because omental tumor-associated CLSs in advanced-stage HGSOC creates a more heterogeneous tumor environment than CLSs induced by obesity, we will perform more studies to clarify the possible mechanisms by which omental CLSs contribute to a poor outcome in advanced

HGSOC. More studies are needed to validate the clinical use of omental CLSs as a prognostic predictor in advanced-stage HGSOC.

**Supplementary Materials:** The following are available online at https://www.mdpi.com/article/10.3390/curroncol28050359/s1, Figure S1: The number of omental CD68+ CLSs exhibited a positive correlation with omental CD163+CLSs; the density of CD68+ om-TAMs had a positive correlation with the density of CD163+om-TAMs. There was little correlation between the density of CD68+ om-TAMs and CD68+ CLSs or between the density of CD163+om-TAMs and the number of CD163+ CLSs; Figure S2: There was a high degree of multicollinearity for CD68+ CLSs and CD163+ CLSs in the model for OS and PFS; Figure S3: In advanced-stage HGSOC, the ovarian tumor density of CD68+ or CD163+ ov-TAMs was not associated with patient prognosis; Figure S4: Representative image of CLS and TAMs by immunohistochemical staining for CD163 and ImageJ software-assisted images; Figure S5: Ten cases for different counts of CLSs existence in omentum after CD68 or CD163 staining; Table S1: Association between clinical parameters and primary ovarian tumor TAM in 116 advanced stage serous ovarian cancer patients.

**Author Contributions:** Conception and design: Y.-L.L., T.-M.H. and K.-F.H. Development of methodology: Y.-L.L., H.-F.T., P.-Y.W. and K.-F.H. Acquisition of data (provided animals, acquired and managed patient data, provided facilities, etc.): Y.-L.L. and K.-F.H. Analysis and interpretation of data (e.g., statistical analysis, biostatistics, computational analysis): Y.-L.L. and S.-H.L. Writing, review, and/or revision of the manuscript: C.-N.L., Y.-L.L. and K.-F.H. Administrative, technical, or material support (i.e., reporting or organizing data, constructing databases): C.-N.L., Y.-L.L., H.-F.T. and P.-Y.W. Study supervision: T.-M.H. and K.-F.H. All authors have read and agreed to the published version of the manuscript.

**Funding:** This research received no external funding.

**Institutional Review Board Statement:** The study was conducted according to the guidelines of the Declaration of Helsinki, and approved by the Institutional Review Board of National Cheng Kun~ University Hospital (protocol code –/I A-ER-108-333).

**Informed Consent Statement:** Informed consent was obtained from all subjects involved in the study.

**Data Availability Statement:** The data presented in this study are available on request from the corresponding author. The data are not publicly available due to patients' privacy.

**Acknowledgments:** This work was supported by grants from the Ministry of Science and Technology of Taiwan (MOST 109–2314-B-006–040), partly from the Ministry of Health and Welfare (MOHW110-TDU-B-211–144018) and the National Health Research Institutes (NHRI-110A1-CACO-02212111). We are also grateful to the Biostatistics Consulting Center, National Cheng Kung University Hospital, for providing the statistical consulting services.

**Conflicts of Interest:** The authors declare no conflict of interest.

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
