# Peer review of "Omental Macrophagic “Crown-like Structures” Are Associated with Poor Prognosis in Advanced-Stage Serous Ovarian Cancer"

_curroncol, doi:10.3390/curroncol28050359_

Round 1
Reviewer 1 Report
The main question addressed by the research is to determine if the presence of CLSs in the omentum correlates with advanced-stage high-grade serous ovarian carcinoma (HGSOC) to be use as a prognostic marker. The topic is not totally in the field of cancer but considerable relevant in ovarian cancer, considering that the omentum is the main site of ovarian cancer metastasis describing these structures is important for the field.
Similar studies have been published for breast cancer; however, it has not been described in omentum and in the context of ovarian cancer contribution is novel for the ovarian cancer field but is not robust because the lack of strong histological evidence in the study.
1. The focus of the paper written in the title are the crown-like structures and the phenotype of the macrophages yet only one figure showing the histology is shown. The CD163 stain is not clear. Figure 1 H and I image shown macrophage staining in what is like a tumor area not a crown-like structure. Even if representative images were chosen for the figure more images showing crown-like structures and macrophage stages could be shown in supplementary figures.
2. Even in the abstract authors write (line 30 to 33) “Conversely, the density of CD68+ or CD163+ TAMs in ovarian tumor was not associated with patient prognosis in advanced-stage HGSOC. In conclusion, we for the first time demonstrate that the presence of omental CLSs is associated with poor prognosis in advanced-stage HGSOC”. Therefore, is not clear if the macrophages are or not part of the prognosis or if is just the crown-like structure regardless of the macrophage phenotype? Then, more histology showing the diverse crown-like structurers would be recommended.
3. Tables and graphs are well done however they are representing counts from histology and figure representing histology is not clear.
Reviewer 2 Report
The Authors report on a novel prognostic factor in ovarian cancer. The statistical analysis is appropriate. No special consideration. Good paper worth to be published after revision of the English form.
